# Microwave Synthesis of Poly(Acrylic) Acid-Coated Magnetic Nanoparticles as Draw Solutes in Forward Osmosis

**DOI:** 10.3390/ma16114138

**Published:** 2023-06-01

**Authors:** Sabina Vohl, Irena Ban, Miha Drofenik, Hermina Buksek, Sašo Gyergyek, Irena Petrinic, Claus Hélix-Nielsen, Janja Stergar

**Affiliations:** 1Faculty of Chemistry and Chemical Engineering, University of Maribor, Smetanova 17, 2000 Maribor, Slovenia; sabina.vohl@um.si (S.V.); irena.ban@um.si (I.B.); miha.drofenik@um.si (M.D.); hermina.buksek@um.si (H.B.); irena.petrinic@um.si (I.P.); clhe@dtu.dk (C.H.-N.); 2Jožef Stefan Institute, Department of Materials Synthesis, Jamova cesta 29, 1000 Ljubljana, Slovenia; saso.gyergyek@ijs.si; 3Department of Environmental and Resource Engineering, Technical University of Denmark, Miljøvej 113, 2800 Kgs. Lyngby, Denmark

**Keywords:** magnetic nanoparticles, microwave synthesis, polyacrylic acid, osmotic pressure, draw solution, forward osmosis

## Abstract

Polyacrylic acid (PAA)-coated magnetic nanoparticles (MNP@PAA) were synthesized and evaluated as draw solutes in the forward osmosis (FO) process. MNP@PAA were synthesized by microwave irradiation and chemical co-precipitation from aqueous solutions of Fe^2+^ and Fe^3+^ salts. The results showed that the synthesized MNPs have spherical shapes of maghemite Fe_2_O_3_ and superparamagnetic properties, which allow draw solution (DS) recovery using an external magnetic field. Synthesized MNP, coated with PAA, yielded an osmotic pressure of ~12.8 bar at a 0.7% concentration, resulting in an initial water flux of 8.1 LMH. The MNP@PAA particles were captured by an external magnetic field, rinsed in ethanol, and re-concentrated as DS in repetitive FO experiments with deionized water as a feed solution (FS). The osmotic pressure of the re-concentrated DS was 4.1 bar at a 0.35% concentration, resulting in an initial water flux of 2.1 LMH. Taken together, the results show the feasibility of using MNP@PAA particles as draw solutes.

## 1. Introduction

In the last decades, the need for drinking water is increasing and becoming a serious global problem. Due to the fast population growth, and very fast industrialization, the ability to reuse wastewater is becoming more and more important. Each year, 3.5 million people die due to a lack of water [1]. In addition, the occurrence of persistent chemicals of emerging concern in potable water remains a very big threat to human health [2]. Much research has been devoted to comparing the efficiency of various water treatment technologies. Classical, albeit energy-intensive, technologies encompass desalination methods including reverse osmosis (RO) and multistage flash distillation [3].

Among the emerging technologies, forward osmosis (FO) is showing promise in product concentration and water extraction. FO, also called direct osmosis, is an evolving technology for membrane separation used in water treatment and reclamation. FO water treatment systems consist of three main components: draw solution (DS), FS, and a selectively permeable membrane. The membrane is positioned between the FS and DS, effectively separating the two. In this setup, water molecules migrate from the FS to the DS side, with the FO process gradually decreasing as the osmotic pressure decreases. At the same time, the solutes migrate from the DS to the FS side, which is an unwanted but inevitable transport direction of matter. The process remains relatively stable until the osmotic pressure on both sides reaches equilibrium [4]. FO uses the pressure differential between a FS and a DS to drive transport from the FS to the DS across a semipermeable membrane. In water extraction, the FS can be an impaired water steam (e.g., brackish water or wastewater) with low osmotic pressure compared to the DS [5]. FO is showing great promise, particularly for the treatment of hypersaline, high fouling, or otherwise challenging feed waters. Unlike pressure-driven membrane processes such as RO, where the feed water is pumped at a high enough pressure to overcome the osmotic pressure differential between the feed and permeate, the difference in osmotic pressure between the feed water and a more concentrated DS drives the filtering process with FO. Consequently, the first filtering phase uses less energy and experiences less fouling and scaling. Following cleaning procedures, better fouling reversibility is also seen. The ultimate result of FO, in contrast to other membrane processes, is a diluted DS rather than purified water. Therefore, a second separation stage is required to both re-concentrate the DS for reuse and to generate a purified water product, unless the diluted DS is beneficial in and of itself or the process is only being run to dewater the feed rather than produce a useful water product [6]. FO is a multipurpose technology applied in many fields, such as desalination [7], power generation [8], wastewater treatment [9], food processing [10,11,12], algae biomass dewatering [13], and sludge treatment [14]. However, FO cannot be utilized as a stand-alone system where water is moved over a membrane by a DS with high osmotic pressure. The DS must be recovered or changed when it becomes diluted. For expanding the use of FO, clever DS and efficient DS recovery techniques are still essential [15]. FO offers several key advantages that position it as an attractive and efficient technology. Among the main advantages of FO are low energy consumption, enabling an energy-efficient process that has significant positive impacts on sustainability, and operating cost reduction [16,17]. Furthermore, FO enables a high percentage of water recovery, which is crucial in the context of a sustainable water supply in water-scarce environments. Another advantage of FO is its ability to reduce the membrane fouling propensity. Due to the use of smaller molecular passages compared to other membrane-based methods, the risk of fouling and sediment formation is decreased. This means that membrane maintenance and cleaning are reduced, resulting in an extended lifespan and a higher system efficiency. Moreover, the design of FO systems is relatively simple and adaptable. With reduced pretreatment requirements and greater flexibility in utilizing different water sources, more efficient and less complex system solutions can be achieved. This adaptability and design simplicity contribute to lower installation and maintenance costs of FO devices. The combination of low energy consumption, cost-effective operation, high water recovery, reduced membrane fouling propensity, and a simple and adaptable system design positions FO as a promising technology for advanced water treatment [5,17,18,19,20]. The effectiveness of the FO process in water production heavily relies on the properties of the DS. The DS plays a crucial role as it needs to exhibit adequate osmotic pressure while also being efficiently recoverable. A key challenge lies in the re-concentration of the DS, which is necessary for the separation process and the production of clean water [21].

The choice of an appropriate DS is of the utmost significance since the DS’s role in the FO process is crucial in regulating both the water flow through the membrane and the expenses associated with regeneration. A number of criteria must be met in order to select an efficient DS. In order to effectively drive the FO process, it must be able to: (i) create a high enough osmotic pressure, (ii) have a low viscosity that facilitates easy pumping throughout the system and enhanced water fluxes, (iii) have a low reverse solute flow, (iv) have a high diffusion coefficient that lowers the internal concentration polarization (ICP), (v) be readily accessible in large quantities at a reasonable price, (vi) be affordable and simple to re-concentrate, and (vii) the toxicity of the DS will be a key worry if there is a chance that the water in the finished product will be contaminated [6].

DSs can be categorized according to the nature of the solute. Generally, three main categories can be highlighted: organic-based compounds, inorganic-based compounds, and synthetic materials [22]. Based on the solute categorization, the following DSs were investigated: gas and volatile compounds, inorganic solutes, organic solutes (fertilizers), simple organics, amphiphilic organic ionic liquids, switchable polarity solvents, organic ionic salts, polyelectrolyte DSs, pH-responsive polymers, thermo-responsive copolymer, hydro-acid complexes, stimuli-responsive hydrogels, MNPs functionalized with simple polymers, quantum dots, and stimuli-responsive nanoparticles [6]. Extensive research has been conducted on various types of DS in FO. Among these, gas-phase compounds, organic solutes, inorganic solutes, and hybrid organic–inorganic nanoparticles have emerged as the most extensively studied options. Each of these DS types offer distinct advantages and challenges. Gas-phase compounds, such as ammonia or carbon dioxide, have attracted attention due to their high osmotic potential and easy recovery through gaseous separation techniques. However, their application in FO systems requires careful consideration of gas solubility, handling, and potential environmental impacts [23]. Organic solutes, including sugars, alcohols, or polymers, have demonstrated favorable osmotic properties and the ability for efficient re-concentration. They offer a wide range of options for customization and optimization to suit specific water treatment requirements. Nonetheless, the selection of organic solutes should consider factors such as solute leakage, potential fouling, and the availability of cost-effective recovery processes [24].

Inorganic solutes, such as salts or brines, have been explored for their high osmotic pressure and abundant availability. They present opportunities for utilizing waste or saline water streams as DSs. However, challenges related to DS recovery, potential scaling, and the impact on membrane performance must be addressed to ensure the viability and sustainability of FO systems [25]. Hybrid organic–inorganic nanoparticles have recently gained attention as innovative draw solutes [26]. These nanoparticles combine the advantages of both organic and inorganic components, offering unique properties such as tunable osmotic pressure, enhanced stability, and potential functionalization for specific applications. However, further research is needed to optimize the synthesis, recovery, and potential environmental impacts of these novel DS candidates [26,27,28,29]. Recently, magnetic nanoparticles (MNPs) have received attention due to their easy surface modifiability, magnetic properties, and biocompatibility. A significant advantage of MNPs compared to other DSs is their easy regeneration using an external magnetic field, provided that the MNPs in ferrofluids do not aggregate. Several papers report how to prevent aggregation between MNPs and how stable ferrofluids can be prepared by the adsorption of surfactants on the MNP particle surface [30,31,32,33]. Recently, MNPs have been proposed as a potential alternative to traditional DSs in FO. The use of MNPs as DSs in FO offers several advantages, including faster separation, high water flux, reusability, lower energy requirements, and enhanced selectivity [32]. However, the colloidal stability of magnetic fluids developed for use as DSs remains a challenge. MNPs can agglomerate or aggregate in aqueous media with varying compositions, including pH, salt concentration, presence of specific ions, or even microbiological activity, which can lead to a reduction in water flux after several cycles. For suspensions of MNPs used as DS in the FO process, it is a requirement that the DS osmotic pressure is higher than for the FS. This can be achieved by surface functionalization with hydrophilicity polymers [34]. In our previous work [35], we reported a two-pot synthesis of magnetite nanoparticles. In the first step, we synthesized magnetite MNPs by precipitation, and the MNPs were determined to be about 13 nm in size. In the second step, the MNPs were coated with (3-aminopropyl)triethoxysilane (APTES), which served as a precursor for the polyacrylic acid (PAA) coating. We used 3-(3-dimethylaminopropyl)carbodiimide (EDC) as the crosslinker, which formed a strong covalent bond (peptide bond) between APTES and PAA. The resulting MNP@APTES@PAA nanocomposite exhibited good colloidal stability in an aqueous solution with an osmotic pressure of 9.19 bar at a 0.59% concentration. The results from FO showed the recovery of the MNP@APTES@PAA composite particles as DS, which confirmed our prediction about the strong covalent bond (peptide bond).

Using a two-pot synthesis, we succeeded in obtaining MNP@APTES@PAA composite particles with very strong covalent bonding. The process was complicated by the use of the EDC crosslinker to form a strong covalent bond to the PAA. To alleviate this, we aimed at finding a faster one-pot synthesis to produce MNP@PAA nanocomposite particles for use as DS in FO. Synthesis of direct covalent PAA coating of MNPs is also faster, cheaper, and environmentally friendlier in comparison with more-pot synthesis. PAA modification has a big influence on MNP characteristics. After surface modification, the nanoparticles’ dispersibility was improved, and the zeta potential of the nanoparticles decreased [36]. PAA is a synthetic polymer with a high molecular weight. It is non-toxic and contains carboxyl groups in each unit. PAA acts as a weak polyelectrolyte, and its degree of dissociation depends on pH and ionic strength. Due to its solubility in water and high density of reactive functional groups, PAA is widely used in applications involving FO. In order to overcome the challenge of MNP agglomeration, considerable efforts have been made to achieve the desired stability and dispersion of MNPs [32,37].

Specifically, we investigated microwave-assisted synthesis relying on efficient energy conversion and uniform heat distribution in the reaction system [38]. In recent years, microwave synthesis has emerged as a promising and environmentally friendly method for the large-scale synthesis of nanomaterials, including MNPs. This innovative synthesis process offers several advantages over conventional methods. Microwave synthesis is known for its simplicity, time-saving nature, and low energy consumption, making it an attractive option for the efficient production of nanomaterials [39,40]. One notable benefit of microwave synthesis is its ability to yield nanoparticles with a narrow size distribution. The controlled and rapid heating provided by microwave irradiation promotes homogeneous nucleation and growth, resulting in nanoparticles with uniform sizes. This narrow size distribution is crucial for achieving a consistent and predictable performance in various applications, including FO. Moreover, microwave synthesis facilitates the formation of nanoparticles with high crystallinity. The rapid and efficient heating provided by microwaves promotes the crystallization process, leading to well-defined and highly crystalline nanoparticles. These crystalline structures contribute to enhanced material properties, such as magnetic behavior, stability, and catalytic activity, which are advantageous for their use in FO applications. Additionally, the nanoparticles synthesized through microwave irradiation exhibit high water solubility. The rapid and efficient synthesis process promotes the formation of surface-functionalized nanoparticles that readily disperse and dissolve in water. This high water solubility is essential for the successful integration of MNPs into the DS used in FO processes, ensuring their effective performance [38,41]. Comparative studies between conventional synthesis methods and microwave-assisted synthesis have demonstrated the superiority of microwave irradiation in terms of reaction rates and yields. The controlled heating and efficient energy transfer in microwave-assisted reactions result in accelerated reaction kinetics and higher product yields. This advantage of microwave synthesis contributes to increased productivity and cost-effectiveness, making it a favorable approach for large-scale production of MNPs. Here, we present a fast method for preparing functionalized MNPs for use as DS in the FO process. The MNPs prepared by this one-pot synthesis method are uniform in size and shape and water-soluble. Stability is ensured by the fact that the PAA chemically bonded to the surface of the MNP, causing both electrostatic and steric repulsion, preventing aggregation. MNPs synthesized with PAA maintain a high osmotic pressure due to the presence of carboxyl groups. This enables efficient water transfer through the membrane during osmosis, thereby enhancing the effectiveness and capacity of FO. Direct covalent PAA bonded on bare MNPs enables the reproducibility of MNP@PAA as DS [35].

## 2. Materials and Methods

### 2.1. Particle Synthesis and Coating

All chemicals were analytical-grade reagents and were used without further purification. Iron (III) chloride (FeCl_3_ × 6H_2_O) was purchased from Carlo Erba reagents GmbH, Germany, while the iron (II) sulphate (FeSO_4_ × 7H_2_O) was purchased from Acros Organics, Thermo Scientific ^TM^ products, USA. Sodium carbonate hydrate (Na_2_CO_3_ × H_2_O) was purchased from Kemika, Croatia, and polyacrylic acid (C_3_H_4_O_2_)_n_ from Sigma Aldrich, Corp. St. Louis, MO, USA. Ethanol (C_2_H_5_OH) from Sigma Aldrich, Corp., St. Louis, MO, USA was used as a washing material. Deionized water (DI) was used for all reactions.

Microwave synthesis experiments were performed in a microwave oven (power changing from 3 W to 6.5 W during synthesis; magneton frequency: 2455 MHz), Discover SP, CEM, USA. In detail: In a 50 mL flask, 0.09 M FeCl_3_ × 6H_2_O and 0.1 M FeSO_4_ × 7H_2_O were dissolved. After the salts were completely dissolved, the solution was poured into a 100 mL microwave flask. The microwave flask containing the dissolved salts was then stirred and heated at 60 °C for 10 s. Then, 8000 μL of 1 M Na_2_CO_3_ was added to the solution and heated at 60 °C for 1 h. After one hour, 0.325 g of PAA (*M*_w_ = 1800 g/mol) was added, and the reaction mixture was stirred at 60 °C for another hour. The product was then separated and prepared for characterization. The MNPs were washed five times with DI water and once with ethanol. The product was redispersed in DI water or dried in an oven at 80 °C for analysis.

### 2.2. X-ray Diffraction Analysis (XRD)

X-ray diffraction measurements with monochromatic CuKα radiation were performed to investigate the crystal structure of MNPs. X-ray diffraction patterns were recorded using a BRUKER D2 PHASER 2ndGen, Karlsruhe, Germany. The patterns were recorded in the range of 20 to 70° (2*Θ*) with a step size of 0.02° 2*Θ* and a rate of 30 s/step. All measured samples were dried before the measurements.

### 2.3. Transmission Electron Microscopy (TEM)

Transmission electron microscopy (TEM) was performed using a JEOL JEM-210 TEM, Tokyo, Japan (thermionic source operated at 200 kV). The colloid sample was dropped onto a holey carbon-coated Cu grid and left to naturally dry in an atmosphere of air. The particle size and size distribution were then determined using custom particle imaging software 2.11, Digital Micrograph Gatan Inc (Pleasanton, CA, USA).

### 2.4. Fourier Transform Infrared Spectroscopy (FTIR)

Fourier transform infrared spectroscopy (FTIR) spectra were measured using an IRAffinity 1S Shimadzu FTIR spectrometer (Pekin-Elmer 5000 Inc., Beaconsfield, UK) in a scan range of 4000–400 cm^−1^. For all measurements, a total of 260 scans were performed with a resolution of 4 cm^−1^. All measured samples were dried before measurement.

### 2.5. Thermogravimetric Analysis (TGA)

Thermogravimetric analysis (TGA) was measured using the TGA 2, Mettler Toledo, Switzerland. Measurements of dried samples were performed under an N_2_ atmosphere with a gas flow of 100 mL/min between 30 and 600 °C at a heating rate of 10 K/min. The weight loss was attributed to the PAA coating of the MNPs.

### 2.6. Dynamic Light Scattering

Particle size and zeta potential were measured using a Zetasizer Nano ZS, Malvern, UK, operated with a 4 mW He-Ne laser at 633 nm. Analyses were performed on dilute MNP@PAA suspensions, so multiple scattering was considered negligible. The isoelectric points were determined by titration and measurement of the respective zeta potentials of the samples.

### 2.7. Magnetic Measurements

Magnetization parameters were measured using a LakeShore 7304 vibration sample magnetometer (VSM), Westerville, OH, USA. The magnetization curves were determined at room temperature.

### 2.8. Osmotic Pressure

Osmolality was determined using a freezing-point osmometer, Gonotec-Osmomat 030, Berlin, Germany. Osmolality was determined for the dispersions prepared from MNP@PAA and then the osmotic pressure was calculated using the following equation in [42].

### 2.9. FO Filtration Experiments

FO filtration experiments were performed using an AIM™ HFFO module with an effective membrane area of 180 cm^2^, kindly provided by Aquaporin A/S, Kgs, Lyngby, Denmark. The batch experiments were performed in cross-flow mode using a double-headed peristaltic pump (Longer Pump^®^ BT 100–1, Shijiazhuang, China), with the FS and DS circulation of 120.1 mL·min^−1^ in counter-current mode. The active side of the membrane was facing the FS. The mass changes of the FS were continuously monitored every 30 s by digital balance (Ohaus Scout Pro, NJ, USA). The specification of the AIM™ HFFO module is presented in a previous paper. The experimental FO setup used in the study is presented in Figure 1.

## 3. Results and Discussion

### 3.1. Synthesis and Particle Size Distribution

The XRD pattern in Figure 2 shows that the synthesized MNPs have a face-centered cubic (fcc) and an orthorhombic crystal phase by comparing with the data from the JCPDS file (00-004-0755) and the JCPDS file (00-029-0713), indicating the formation of maghemite and goethite. The peaks at 2*Θ* = 30.2°, 35.5°, 57.1°, and 62.9° can be indexed to the (220), (311), (400), and (511) lattice planes of cubic maghemite, and the peaks at 2*Θ* = 21.3°, 33.2°, 36.9°, and 53.2° can be indexed to the (111), (200), (220), and (311) lattice planes of orthorhombic goethite. XRD analysis showed that the maghemite content was about 90%, while the goethite content was about 10%. The particle size estimated from the above peaks yielded an average value of 7 nm for maghemite and 9.5 nm for goethite, based on the Scherrer equation. No impurities from other iron oxides were observed. The strong magnetism can be seen from the inset pictures in Figure 2. The MNPs rapidly responded to an external magnetic field. The MNPs can be sedimented from their suspension by being attracted to a magnet, and they can be redispersed in water after the magnet is removed. This property facilitates the separation of MNPs during their preparation.

Observation of MNP@PAA using TEM (Figure 3a) revealed that the maghemite nanoparticles were of uniform size, with an average size *d*_TEM_ = 7.0 nm ± 1.9 nm (Figure 3 inset), which was in good agreement with the average size determined from the broadening of X-ray lines. Observation at a higher magnification (Figure 3b) showed that individual maghemite nanoparticles were single crystalline.

The saturated mass magnetization was determined using a vibrating sample magnetometer. Figure 4 shows the magnetization measured at room temperature for the MNP@PAA prepared with the microwave oven. The MNP@PAA exhibited typical superparamagnetic behavior at room temperature with a saturation magnetization of 19.4 emu/g. The saturation magnetization of the as-prepared nanoparticles was 65 emu/g [43]. When the MNPs were coated with the selected coating, it can be observed that the magnetization decreased. As can be seen in Figure 4, the magnetization showed no remanence and coercivity.

### 3.2. Surface Characterization

MNPs were directly coated in the microwave flask by injecting PAA. This one-pot synthesis and coating have the advantage of eliminating time-consuming washing and cleaning steps common to conventional methods. The success of the coating was determined by FTIR spectral analysis. Figure 5a shows the FTIR spectrum of bare MNP. For the typical spectrum of goethite, the main peaks at 3149, 886, and 791 cm^−1^ were assigned to the vibrations of -OH. The peak around 582 cm^−1^ showed Fe-O bonding, which is typical of maghemite [44,45]. If we compare the synthesis result with the XRD analysis, we can also confirm the formation of goethite and maghemite with the FTIR analysis. Figure 5b shows the FTIR spectrum of MNP@PAA (red) and pure PAA (black). For PAA, the main peaks at 2949, 1695, 1446, and 1417 cm^−1^ were assigned to -CH_2_- (stretching and bending), -COO (stretching in -COOH), and C-O (stretching in -COOH), respectively [31]. The FTIR spectrum of MNP@PAA showed that the peak at 1695 cm^−1^ shrank and a new peak appeared at about 1450 cm^−1^, which was due to the binding of carboxylic acid groups to the surface of MNPs to form carboxylate groups. The new peak corresponded to the COO^−^ vibration, which indicates the bonding of the carbonyl groups to the surface Fe atoms [31,46].

Figure 6 shows the TGA of bare MNPs, the MNP@PAA synthesized in a microwave oven, and the PAA (*M*_w_ = 1800). Weight loss below 200 °C can be attributed to the removal of bound water (Figure 6a,b). After 200 °C, there was a mass loss of around 12% between 200 and 550 °C, and this was due to the conversion of goethite and maghemite to hematite (Figure 6a, black curve). In Figure 6b (red curve), we can see a mass loss of around 24%, and this was due to PAA on the surface of nanoparticles. The mass loss in the interval between 270 °C and 600 °C was attributed to the decomposition of the PAA polymer chains [47]. The main degradation temperature of the PAA was near 400 °C [31].

In addition to FTIR and TGA characterization of the particle surface, the change in the isoelectric point of the particle sample can be used to determine the success of the surface modifications. The pH-dependent zeta potential of MNP@PAA was measured in a titration study using HCl and NaOH. The pH measurement ranged from 2 to 9. At neutral pH, the MNP@PAA were very negatively charged at about −35 mV. The negative zeta potential indicated that the PAA was bound to the particle surface. When the solution became more alkaline, the zeta potential became more negative, which could be due to the ionization of PAA [31]. The ionization of PAA would cause the electrostatic repulsion against the aggregation between MNPs. The isoelectric point for MNP@PAA was at pH 3.55 (Figure 7), which indicates that the MNPs were coated with PAA [35].

### 3.3. Results of FO Experiments

Two FO filtrations were performed using a suspension of MNP@PAA as a DS and DI water as an FS. In Figure 8, the results of the water flux versus time are presented, while in Figure 9a–d, the results of the conductivity of the FS and DS, the osmotic pressure of the FS and DS, and the pH value of the FS and DS are presented, respectively, versus time.

For the first filtration (blue curves in Figure 8 and Figure 9), we used 190 mL of a freshly synthesized 0.70% suspension of MNP@PAA with an osmotic pressure of 13.0 bar and 500 mL of DI water as an FS. During the first filtration, the water flux decreased from a maximum value of 8.1 LMH to 3.4 LMH in two hours. During that time, the membrane passed through 121.9 mL of water (water recovery was 24.5%), which corresponds to an increase in the dilution factor of FS from 1 to 1.6. The osmotic pressure of the FS after two hours increased to 2.45 bar, while an increment in conductivity from 11.0 μS cm^−1^ to 13.4 μS cm^−1^ was noticed. For DS, the conductivity values were scattered, and a slight decrease was observed over the whole filtration time, from an average value of 28 μS cm^−1^ to 22.7 μS cm^−1^ ± 8. μS cm^−1^. The osmotic pressure difference between DS and FS at the end of filtration was 2.0 bar. The pH value of the FS was 6.9 at the beginning of the filtration and 6.8 at the end, and for the DS the initial pH value was 5.1, and the final value was 5.3. After the first filtration, 310 mL of diluted DS (suspension of MNP@PAA) was obtained, which means that about 110 mL of water permeated from FS to the DS. The osmotic pressure of the diluted DS was 4.4 bar, corresponding to a *w*/*w* concentration of 0.35%.

In order to reuse the MNP@PAA from the diluted DS during the first filtration, a permanent magnet was used for separation, followed by a rinsing procedure with ethanol. The particles were then re-suspended. The second filtration (red curves in Figure 8 and Figure 9) was performed using 190 mL of a 0.35% suspension of the re-suspended MNP@PAA with 500 mL of DI water as an FS. The osmotic pressure of the DS at the beginning of FO testing was 4.1 bar. During the second 2 h filtration, the water flux decreased from a value of 2.1 LMH to 1.4 LMH, where 51.1 mL of water permeated the membrane (corresponding to a water recovery of 10.3%), which corresponds to an increase in the dilution factor of FS from 1 to 1.3. The osmotic pressure of the FS after two hours increased to 0.6 bar, while the increment in conductivity was insignificant. For DS, the decrease in conductivity was more pronounced than in the first filtration, over the filtration time, from an average value of 55.2 μS cm^−1^ to 22.7 μS cm^−1^ ± 6.3 μS cm^−1^. The osmotic pressure difference between DS and FS at the end of filtration was 1.0 bar. The pH value of the FS was 6.4 at the beginning and 7.1 at the end. For the DS, the pH value was 5.3 at the beginning and 5.4 at the end of the experiment. After the second filtration, 240 mL of diluted DS (suspension of MNP@PAA) was obtained, which means that about 50 mL of water permeated from FS to the DS. The osmotic pressure of the diluted DS was 1.6 bar, which corresponds to a *w*/*w* concentration of 0.23%.

When comparing the freshly synthesized suspension of MNP@PAA with a suspension of used and magnetically separated MNP@PAA, we observed a decrease in the concentration, and consequently in the osmotic pressure. The starting concentration of 0.70% (first filtration) decreased to 0.35% (second filtration), which was reflected in the osmotic pressure values, i.e., 13.0 bar decreased to 4.1 bar. When comparing the filtration performance, we can see that the starting water flux for the first filtration was 8.1 LMH, while the starting water flux for the second filtration was 2.1 LMH—3.8-times lower compared to the first filtration. The starting osmotic pressure was 3.2-times lower if we compared the starting driving force of the DS for both filtrations.

### 3.4. Characterization after the First and Second FO Processes

After both the first and second filtrations, we examined the particles to see if the PAA remained bound to the MNPs. After both filtrations, the particles were stable in solution (no aggregation) and maintained their pressure, indicating that they were still coated with PAA. To confirm this, the MNP was characterized after both filtrations by TGA, DLS, and FTIR analysis. Using TGA, we determined the mass loss before and after the first and second filtrations. For all measurements and analyses performed after filtration, we used the same conditions as for the characterization of MNP@PAA. The TGA results in Figure 10 show the weight loss of MNPs after the first FO process (MNP@PAA_afterFO1) and after the second FO process (MNP@PAA_afterFO2). The overall weight loss of MNP@PAA before the FO process was determined as 23.47%, after the first FO as 23.83%, and after the second FO process as 22.90%. The mass loss after FO did not significantly change, confirming that PAA remained on the particles during the FO process and was not washed off, indicating a strong covalent bond.

We also characterized the MNPs after the first and second FO processes using DLS, with the same measurement conditions as before the FO process. Particle titration was performed before the FO process and the isoelectric point was determined to be 3.55 (see Figure 7). We also performed titrations after the first and second FO processes and determined the isoelectric point, as shown in Figure 11a,b. After the first FO process, we determined the isoelectric point as 3.60 (Figure 11a), and the isoelectric point after the second FO process was determined as 3.75 (Figure 11b). Again, the values before and after the FO process differed only slightly, and we can argue, similar to the TGA, that the particles were still coated with PAA after FO.

As a third characterization method after the FO process, we chose FTIR analysis, which showed the following results (Figure 12). Figure 12 shows the spectrum after the first (black curve) and after the second FO process (red curve). Comparing the spectra for pure PAA and MNP@PAA before the FO process (Figure 5b), we see that the spectra did not differ. From this, we confirmed that the particles remained coated with PAA after both FO filtrations.

## 4. Conclusions

MNP@PAA were successfully synthesized by microwave irradiation. PAA was directly covalently bonded to the MNP surface without the help of an EDC crosslinker, as in our previous studies on PAA-coated MNPs. The MNP@PAA nanocomposites showed excellent colloidal stability in an aqueous solution, with an osmotic pressure of 12.8 bar (0.7% suspension). To verify the stability of the nanoparticle coating with PAA, systematic analyses were performed using techniques such as dynamic light scattering (DLS), Fourier transform infrared spectroscopy (FTIR), thermogravimetric analysis (TGA), X-ray diffraction (XRD), and transmission electron microscopy (TEM).

TGA, FTIR, and isoelectric point measurements confirmed the hydrophilic surface chemistry of the MNP@PAA nanocomposites. XRD analyses indicated a maghemite crystal structure, while TEM analysis revealed a single crystalline structure with an approximate diameter of 7 nm. The saturated mass magnetization of the MNP@PAA nanocomposites was 19.4 emu/g, while the as-prepared nanoparticles had a saturated mass magnetization of 65 emu/g. The osmotic pressure of MNP@PAA was 12.8 bar (0.7% suspension). Two forward osmosis filtrations (FO) were performed using MNP@PAA nanocomposites as the DS. TGA was performed after each FO filtration, and the results showed that the weight losses of MNP@PAA before and after FO were almost identical (23.83% after the first FO and 22.90% after the second FO). FTIR and isoelectric point measurements also confirmed the presence of the PAA coating on the MNP surface after the FO process. The MNP@PAA nanocomposites exhibited excellent colloidal stability in an aqueous solution with high osmotic pressure. The hydrophilic surface chemistry ensured compatibility with aqueous solutions and contributed to improved dispersion and system stability. These results demonstrated the reproducibility of MNP@PAA nanocomposite particles as a reliable DS.

## Figures and Tables

**Figure 1 materials-16-04138-f001:**
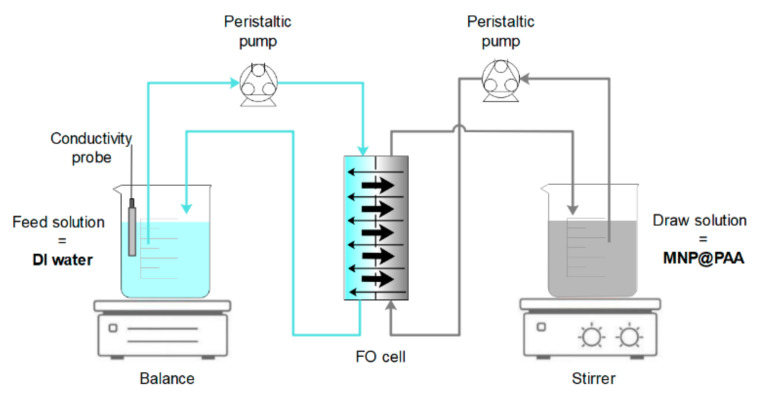
Experimental FO setup.

**Figure 2 materials-16-04138-f002:**
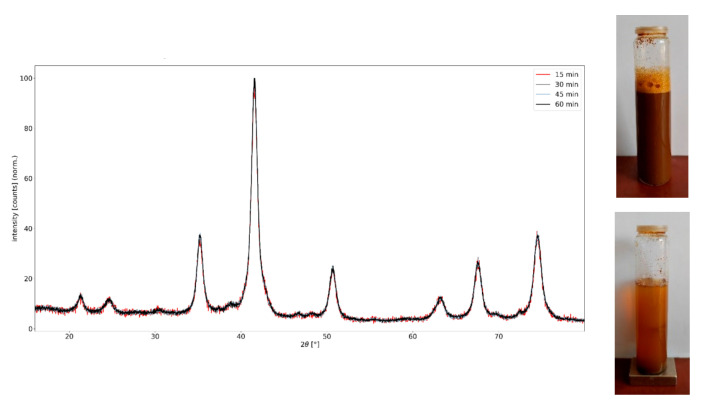
XRD pattern of MNP@PAA synthesized by microwave synthesis.

**Figure 3 materials-16-04138-f003:**
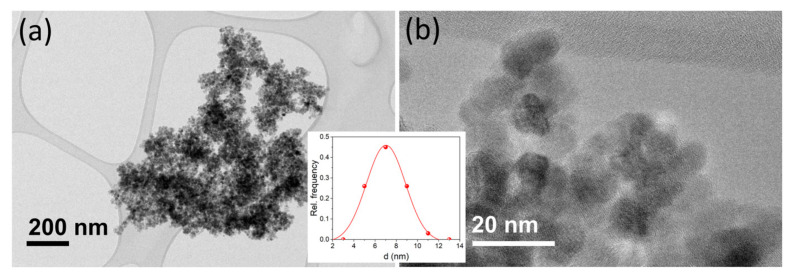
TEM image of MNP@PAA acquired at lower (**a**) and higher (**b**) magnification. The inset is a number-weighed empirical size distribution function (red dots) fitted with a Gaussian function (red line).

**Figure 4 materials-16-04138-f004:**
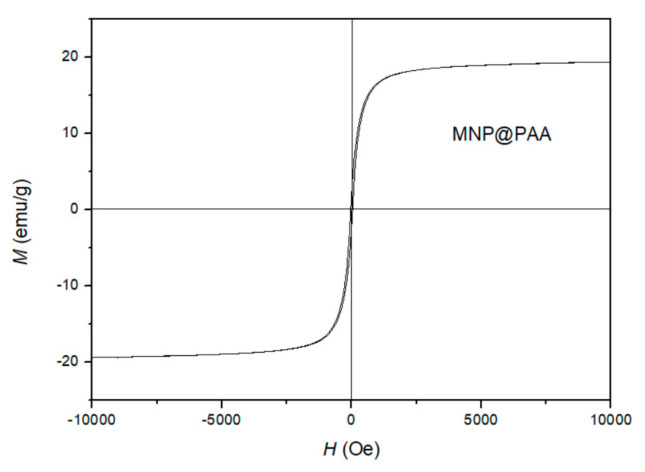
The measured room temperature magnetization curve for MNP@PAA prepared by a microwave oven.

**Figure 5 materials-16-04138-f005:**
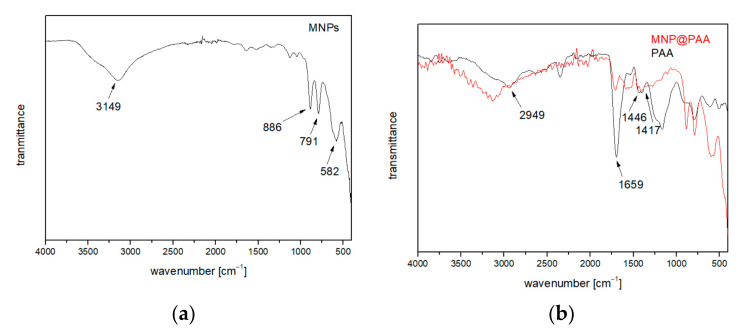
(**a**) FTIR spectra of bare MNP. (**b**) FTIR analysis comparison of MNP@PAA (red curve) and PAA (black curve).

**Figure 6 materials-16-04138-f006:**
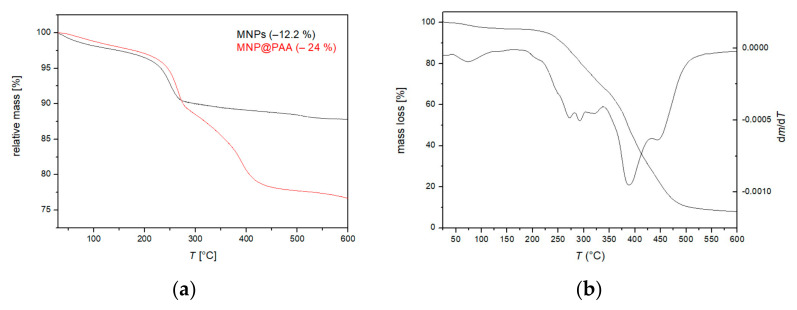
(**a**) TGA of the MNPs and MNP@PAA and (**b**) PAA.

**Figure 7 materials-16-04138-f007:**
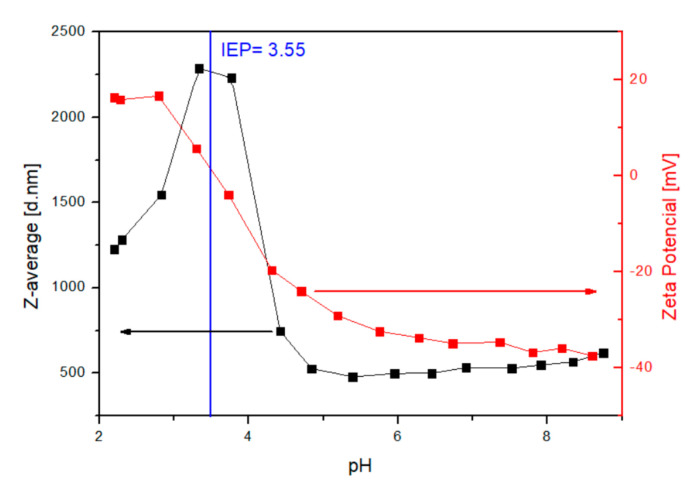
Change of zeta potential (red curve) and hydrodynamic radii (black curve) for MNP@PAA with pH values.

**Figure 8 materials-16-04138-f008:**
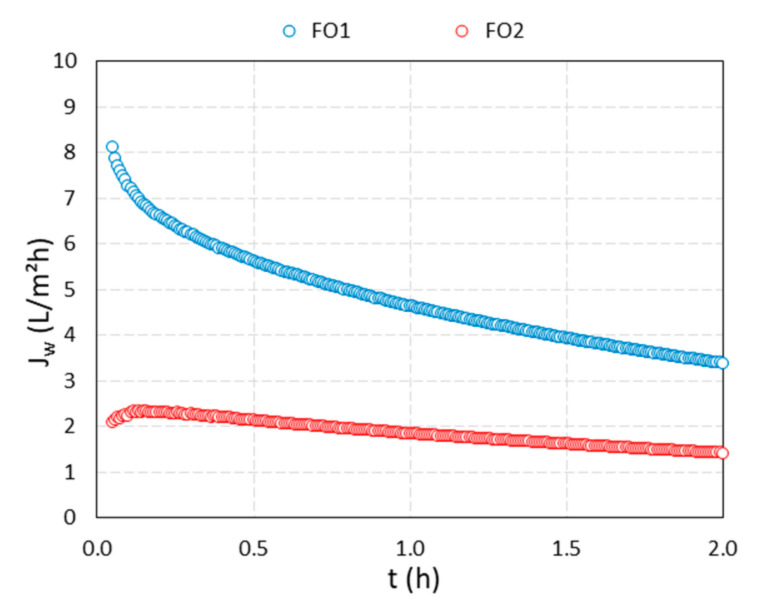
Water flux for 2 h FO filtration test, using freshly synthesized MNP@PAA suspension as a DS (blue curve) and MNP@PAA suspension out of the first filtration (red curve).

**Figure 9 materials-16-04138-f009:**
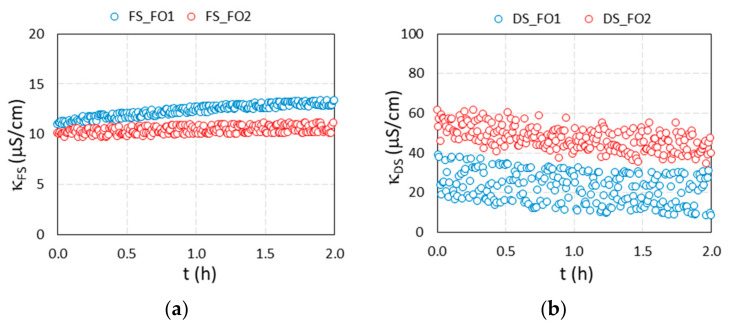
Recorded parameters for the 2 h FO filtration test, using freshly synthesized MNP@PAA suspension as a DS (blue curves) and MNP@PAA suspension out of the first filtration (red curves). (**a**) Conductivity of the FS, (**b**) conductivity of the DS, (**c**) osmotic pressure of the FS and DS, and (**d**) pH value of the FS and DS.

**Figure 10 materials-16-04138-f010:**
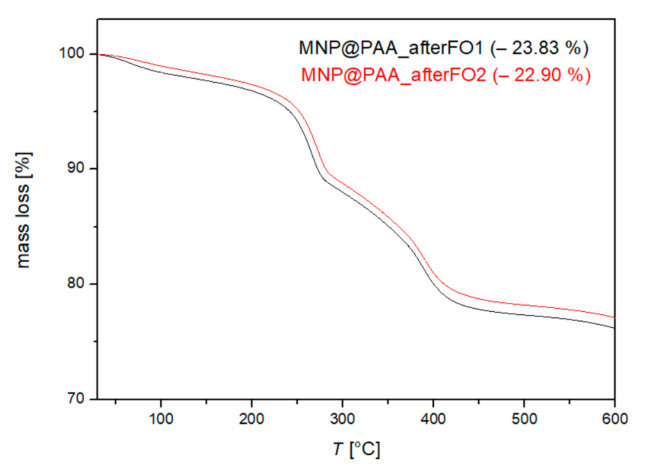
TGA curve of MNP@PAA before and after the first and second FO processes.

**Figure 11 materials-16-04138-f011:**
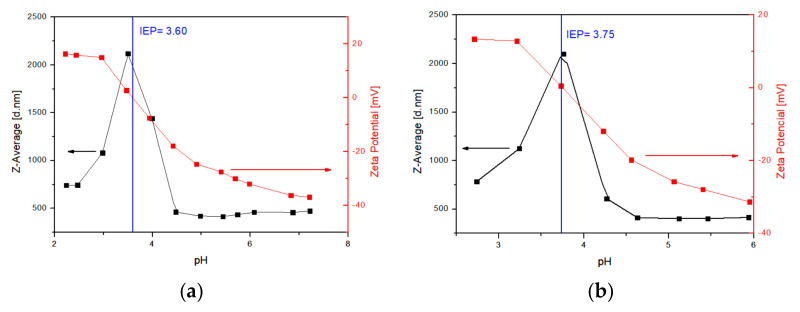
DLS analysis after the first FO process (**a**) and after the second FO process (**b**).

**Figure 12 materials-16-04138-f012:**
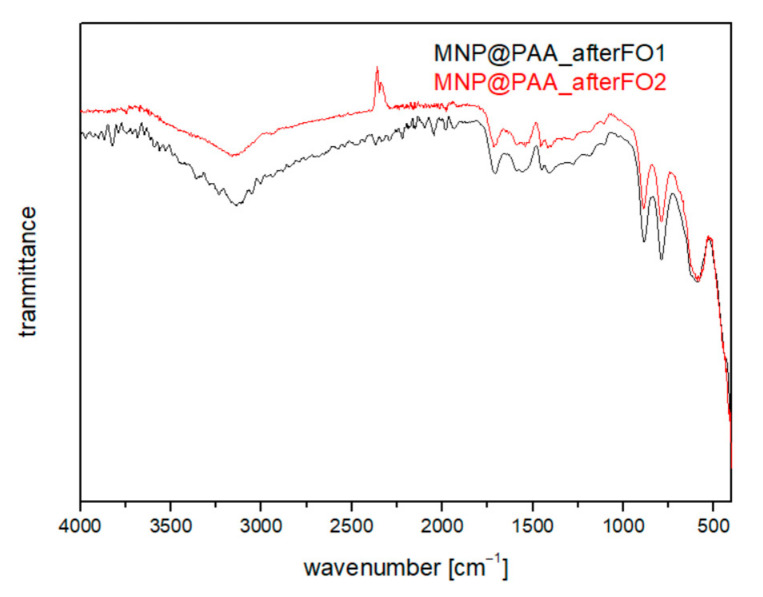
FTIR analysis after the first FO process (black) and after the second FO process (red).

## Data Availability

Not applicable.

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
