# Peer review of "Microwave Synthesis of Poly(Acrylic) Acid-Coated Magnetic Nanoparticles as Draw Solutes in Forward Osmosis"

_materials, 2023, doi:10.3390/ma16114138_

Round 1

Reviewer 1 Report

Recently, magnetic nanoparticles (MNPs) have received attention due to easy surface modifiability, magnetic properties, and biocompatibility. The authors using a one-pot synthesis obtained MNP@PAA composite particles with very strong covalent bonding, for use as DS in FO. The process is complicated by the use of the EDC crosslinker to form a strong covalent bond to the PAA. The microwave synthesis is a promising, large-scale, and green process for the synthesis of nanomaterials. The authors explain that the process is simple, time-saving, and low-energy-consuming, and the synthesized nanoparticles have a narrow size distribution, high crystallinity, and high water solubility [11, 12]. They found that microwave irradiation resulted in higher reaction rates and yields. We present a fast method for preparing functionalized MNPs for use as DS in the FO process. Using a one-pot synthesis obtained MNP@PAA composite- the subject matter is original and suitable for publication. 

New specific improvements: 

1) MNP@PAA are successfully synthesized using microwave irradiation, by PAA directly covalently bonded to the MNP surface without the help of an EDC crosslinker; 

2) MNP@PAA nanocomposites showed good colloidal stability in an aqueous solution with an osmotic pressure of 12.8 bar; 

3) TGA, FTIR, and isoelectric point measurements confirm the hydrophilic surface chemistry of MNP@PAA nanocomposites and TEM analysis reveals a single crystalline structure; 

4) TGA  analyses were performed and the results show that the weight losses of MNP@PAA before and after FO are almost the same; 

5) the reproducibility of MNP@PAA nanocomposite particles as DS. The conclusions are consistent with the evidence and arguments presented and answer the main question posed. References are adequate, but only 3% of them are from the last 3 years - updating is needed to further demonstrate the innovativeness of the proposed research. 

The conclusions are consistent with the evidence and arguments presented and answer the main question posed. References are adequate, but only 3% of them are from the last 3 years - updating is needed to further demonstrate the innovativeness of the proposed research.The discussion of the results is somehow hidden in the description of the results themselves - please update it and add more articles supporting the current moments in the submitted manuscript.

Revise after major revision (lack of control in some experiments) 

Minor editing of English language required

Author Response

Dear Reviewer,

The paper we have written is an original scientific article and covers our work related to the one-pot synthesis of polyacrylic acid- coated magnetic nanoparticles with very strong covalent bonding for use as draw solutes in forward osmosis process. We expanded and improved the introduction and discussion parts, as was suggested. All new comments are marked with "track changes".

Second, we have updated the references to provide additional evidence of the innovative content of our research.

Thank you for your comments, we hope that improved article will meet all requirements and will be ready to publish. If there is anything more left to improve, please don’t hesitate to write us.

Sincerely,

S. Vohl, I. Ban, M. Drofenik, H. Buksek, S. Gyergyek, I. Petrinic, C. Helix-Nielsen and J. Stergar

Reviewer 2 Report

General Assessment:

The manuscript titled "Polyacrylic Acid (PAA)-coated magnetic nanoparticles (MNP@PAA) as draw solutes in forward osmosis (FO)" presents the synthesis and characterization of MNP@PAA and evaluates their potential application as draw solutes in the FO process. The study provides detailed information about the synthesis method, particle characterization, and performance evaluation. While the work demonstrates promising results, some areas require further clarity and additional data to strengthen the manuscript.

Introduction

I recommend expanding the introduction section to provide a more comprehensive background and context for your study. Currently, the introduction provides a brief overview of the research topic but lacks an in-depth discussion on the significance of the work and its relation to existing literature. In addition, expanding this section will allow readers to understand better the motivations behind your study and its novelty in the field of forward osmosis (FO).

Consider including the following points in the expanded introduction:

  1. Present a broader context: Begin by introducing the concept of FO and its potential applications. Then, discuss the advantages and challenges of FO compared to other separation processes, highlighting the need for efficient and sustainable draw solutes.
  2. Literature review: Provide a more comprehensive literature review on draw solutes in FO. Discuss the types of draw solutes investigated, their advantages and limitations, and any notable research gaps or unresolved issues. This will help establish the importance and novelty of your study.
  3. Motivation and objectives: Clearly state the motivation behind your study and the specific objectives you aimed to address. Explain how using polyacrylic acid-coated magnetic nanoparticles as drawn solutes can overcome the limitations of previous approaches and contribute to the field.
  4. Research significance: Highlight the potential impact of your study, both in terms of scientific advancements and practical applications. Discuss how the findings can enhance the efficiency, selectivity, and sustainability of the FO process.

Methods:

5.      The methods section provides sufficient details on the synthesis and coating of MNP@PAA. However, certain aspects could be expanded for better reproducibility, such as the specific microwave conditions (power, frequency) used during the synthesis.

6.      Clarify experimental details: Include specific details about the replicate experiments performed and statistical analyses, if applicable.

Results:

7.      The key results section presents the characterization data obtained for the MNP@PAA particles. The XRD analysis reveals the crystal phases present, and the TEM images confirm the particle size and single crystalline nature. In addition, the FTIR analysis provides insight into the successful coating of PAA on the nanoparticles. However, additional data, such as dynamic light scattering (DLS) measurements to evaluate size distribution and magnetic measurements for saturation magnetization, would enhance the comprehensiveness of the results.

8.      Discuss practical implications: Provide a more comprehensive discussion on the potential applications and advantages of MNP@PAA as draw solutes in FO. Compare the performance of MNP@PAA with other draw solutes, addressing their strengths and limitations.

Conclusion:

9.      The conclusion section summarizes the main findings and highlights the potential of MNP@PAA as draw solutes in FO. It describes the synthesis process, the stability of the nanocomposites, and the osmotic pressure achieved. However, more discussion on the practical implications of these findings, such as the advantages and limitations compared to other draw solutes, would add depth to the conclusion.

10.   Language and Structure: Ensure the manuscript is written clearly and concisely, with a logical flow of information. Proofread the manuscript thoroughly to correct any grammatical or typographical errors.

Overall, this manuscript presents a valuable study on the synthesis and characterization of MNP@PAA as draw solutes in the FO process. Addressing the suggested improvements will enhance the manuscript's scientific rigor and improve its impact in the field of osmotic processes.

Author Response

Dear Reviewer,

We have considered your recommendations regarding the introduction. We have expanded the introduction section with additional literature and in-depth discussion so that the reader can better understand the novelty of the presented study in the field of advanced osmosis. We have attempted to further explain the benefits and challenges of advanced osmosis compared to other separation processes, highlighting the need for efficient and sustainable pumping solutions. We have further justified and explained our motivation for our research and defined goals that explain how the use of polyacrylic acid- coated magnetic nanoparticles can go beyond previous approaches and contribute to the field. In the introduction section, we have additionally explained the significance of our research, and everything mentioned can be traced with the help of "track changes".

Secondly, as you suggested, in the method chapter we added conditions such as power and frequency that were used in the synthesis itself. Regarding the reproducible experiments, magnetic nanoparticles were synthesized several times, and all batches were always characterized, and the results were reproducible.

Regarding the results section, we expanded and improved the discussion as you suggested.

In the conclusion, we summarized the main results and highlighted the potential of polyacrylic acid- coated magnetic nanoparticles as a target solution in forward osmosis. We have included an additional discussion in the introduction and in the results section, where we elaborated the advantages and limitations of our magnetic nanoparticles compared to other magnetic nanoparticles used.

If there is anything more, we need to improve, please don’t hesitate to contact us.

Sincerely,

S. Vohl, I. Ban, M. Drofenik, H. Buksek, S. Gyergyek, I. Petrinic, C. Helix-Nielsen and J. Stergar

Round 2

Reviewer 1 Report

materials-2413881

The introduction, matherials and metods and conclusions are presented against the proposed remarks. References supplements deepen and shape the manuscript.

Note - to be uniform font of the text, according to the rules of the journal!

Minor editing of English language required